# Thermoelectric current in a graphene Cooper pair splitter

Z. B. Tan[1,2], A. Laitinen [1], N. S. Kirsanov [1,3,4,5], A. Galda[6,7], V. M. Vinokur[5,7], M. Haque [1], A. Savin [1], D. S. Golubev[8], G. B. Lesovik[3,4] & P. J. Hakonen [1,8 ✉]

Generation of electric voltage in a conductor by applying a temperature gradient is a fundamental phenomenon called the Seebeck effect. This effect and its inverse is widely exploited in diverse applications ranging from thermoelectric power generators to temperature sensing. Recently, a possibility of thermoelectricity arising from the interplay of the non-local Cooper pair splitting and the elastic co-tunneling in the hybrid normal metal-superconductor-normal metal structures was predicted. Here, we report the observation of the non-local Seebeck effect in a graphene-based Cooper pair splitting device comprising two quantum dots connected to an aluminum superconductor and present a theoretical description of this phenomenon. The observed non-local Seebeck effect offers an efficient tool for producing entangled electrons.

[1] Low Temperature Laboratory, Department of Applied Physics, Aalto University, Espoo, Finland. [2] Shenzhen Institute for Quantum Science and Engineering, Southern University of Science and Technology, Shenzhen 518055, China. [3] Terra Quantum AG, St. Gallerstrasse 16A, 9400 Rorschach, Switzerland. [4] Moscow Institute of Physics and Technology, Institutskii Per. 9, Dolgoprudny, Moscow Distr. 141700, Russian Federation. [5] Consortium for Advanced Science and Engineering (CASE), University of Chicago, 5801 S Ellis Avenue, Chicago, IL 60637, USA. [6] James Franck Institute, University of Chicago, Chicago, IL 60637, USA. [7] Materials Science Division, Argonne National Laboratory, 9700 S. Cass Avenue, Argonne, IL 60439, USA. [8] QTF Centre of Excellence, Department of Applied Physics, Aalto University, FI-00076 Aalto, Finland. ✉email: pertti.hakonen@aalto.fi

**M**esoscopic thermoelectric effects have been investigated in a variety of condensed matter systems that, besides fundamental normal metal–superconductor–normal metal (NSN) systems[1–5], also include quantum dots[6–10], atomic point contacts[11–13], Andreev interferometers[14,15], superconducting rings[16] and nanowire heat engines[17]. Thermoelectric effects in the superconducting systems[18–22], in particular those dealing with non-local thermoelectric currents in superconductor–ferromagnet devices[23–25] and in bulk non-magnetic hybrid NSN structures[26–28] have attracted special attention. The connection between thermoelectric effects and the Cooper pair splitting (CPS)[1,2], proposed in ref. [29], established a mechanism for the coherent non-local thermoelectric effect in hybrid superconducting systems. This connection was further studied and explicitly described for a ballistic NSN structure[4]. It was revealed analytically in ref. [4] that the electric transport in the NSN structures depends on the elastic co-tunneling (EC) process on par with the CPS. Contrary to intuitive expectations, together these two processes may enable the transfer of heat through the superconductor[3–5]. The EC and CPS probabilities can in turn be made energy dependent by placing quantum dots between each normal lead and the superconducting region[1,2].

Here we present the experimental observation of the non-local thermoelectric current generated by imposing thermal gradient across a quantum dot–superconductor–quantum dot (QD-S-QD) splitter. We find that both CPS and EC processes contribute to the non-local thermoelectric current and that their relative contributions can be tuned by the gate potentials. The ability to tune between the CPS and EC allows for testing of fundamental theoretical concepts relating entanglement and heat transport in the graphene CPS systems.

## Results

**Theoretical considerations**. Let us consider an QD-S-QD device within the Landauer formalism. Taking that the non-local transport is primarily coherent and that the electron energies are smaller than the superconducting gap, $|E| < \Delta$, we find, see Supplementary Note 4, that the EC, $\tau_{EC}(E)$, and CPS, $\tau_{CPS}(E)$, probabilities are given by the expressions

$$\tau_{EC} = \tau_L(E)\tau_S\tau_R(E), \quad \tau_{CPS} = \tau_L(E)\tau_S\tau_R(-E). \tag{1}$$

Here $\tau_{L(R)}(E)$ is the transmission probability of the left (right) quantum dot renormalized by Coulomb interaction, which depends on the energy of an electron and on the side gate potentials applied to the dots $V_{sg,L(R)}$ ($\tau_{L(R)}(E)$ is given by the sum of Lorentzian peaks or Fano resonances associated with discrete energy levels, see Supplementary Note 4, and $\tau_S$ is the effective transmission probability of the superconducting lead. The latter corresponds to the probability for an electron coming out of one dot so that, instead of escaping into the bulk of the superconducting electrode, it reaches the other dot. It becomes independent on the electron energy $E$ if the dots are separated by a distance shorter than the superconducting coherence length. This condition is reasonably well fulfilled in our experiment. The non-local thermoelectric currents in the dots can, in turn, be expressed in terms of the EC and CPS contributions, $\Delta I_L^{nl} = (\Delta I_{EC} + \Delta I_{CPS})/2$, $\Delta I_R^{nl} = (-\Delta I_{EC} + \Delta I_{CPS})/2$, where

$$\Delta I_{EC} = \frac{2e}{h}\int dE\, \tau_{EC}(E)[f_L(E) - f_R(E)],$$
$$\Delta I_{CPS} = \frac{2e}{h}\int dE\, \tau_{CPS}(E)[f_L(E) - f_R(E)], \tag{2}$$

and $f_{L(R)}(E) = 1/(1 + e^{E/k_B T_{L(R)}})$ is the distribution function in the left (right) electrode having the temperature $T_{L(R)}$.

**Experiment**. Now, we turn to experimental realization of CPS and EC. Several material platforms have been employed in the experiments[30–37], and particularly promising results have been obtained in carbon nanotube, graphene, and nanowire settings, where the splitting efficiencies approaching 90% have been observed. Our present device, depicted in Fig. 1, consists of an Al superconducting injector in contact with two graphene quantum dots. Two side gate electrodes allow us to tune the resonance levels of the dots independently. In order to perform thermoelectric measurements, our device additionally contains two thermometers and a resistive heater, fabricated from a graphene monolayer. The thermometers are superconductor–graphene–superconductor (SGS) Josephson junctions that reveal local temperature through the temperature dependence of the switching current, $I_{sw}(T)$[38]. The resistive heater comprises the graphene nanoribbon and two attached aluminum leads. The heater is distinctly apart and electrically isolated from the rest of the device, the heat to the Cooper pair splitter being transmitted through the substrate.

The temperature difference $\Delta T = T_L - T_R$ between the leads of the two-terminal device induces the thermoelectric current $I = \alpha G \Delta T$, where $G$ is the conductance of the device and $\alpha$ is the Seebeck coefficient[39]. For typical metals, such as aluminum or copper, the Seebeck coefficient is quite small, $\alpha \sim 3–7\,\mu V/K$. For graphene, $\alpha$ is inversely proportional to the square root of charge density, and it can reach much higher values close to the charge neutrality point[40,41]. In quantum dots with energy-dependent electron transmission probability[42], and in superconductor–ferromagnet tunnel junctions[25] large $\alpha$ up to a few $k_B/e \sim 100\,\mu V/K$ can be achieved. In our experiment, we observe similar values of the Seebeck coefficient in graphene quantum dots. We operate the graphene heater at frequency $f = 2.1\,Hz$ and record thermoelectric currents through both quantum dots at the double frequency $2f$ (see "Methods"). Thermal gradient induced by the heater is measured by SGS thermometers, which were calibrated separately as discussed in Supplementary Note 2.

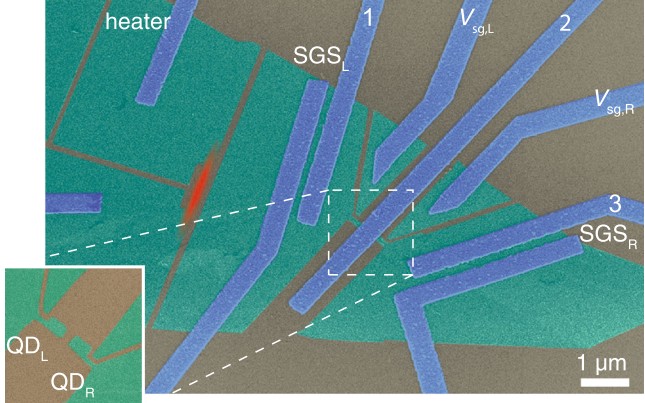

**Fig. 1 False color SEM image of the device.** Green indicates graphene, blue corresponds to metallic Al/Ti sandwich leads, and the silicon substrate with 280-nm-thick silicon dioxide on top is colored in gray. The Joule heated region is indicated by red color. The superconducting graphene junctions are located between the leads marked by SGS$_L$ and SGS$_R$. The left and right graphene quantum dots QD$_L$ and QD$_R$, respectively, have an area $200 \times 150\,nm^2$, foremost located under the Al injector and thus invisible in the image. Side gates with voltages $V_{sg,L}$ and $V_{sg,R}$ are also carved out of graphene. The inset at lower left corner illustrates the graphene quantum dots before overlaying the metallic Cooper pair injector. In the thermoelectric measurements, the quantum dot currents are tracked by current preamplifiers connected to leads 1 and 3 (virtual ground 20 Ω), while the Cooper pair injector, lead 2, is grounded.

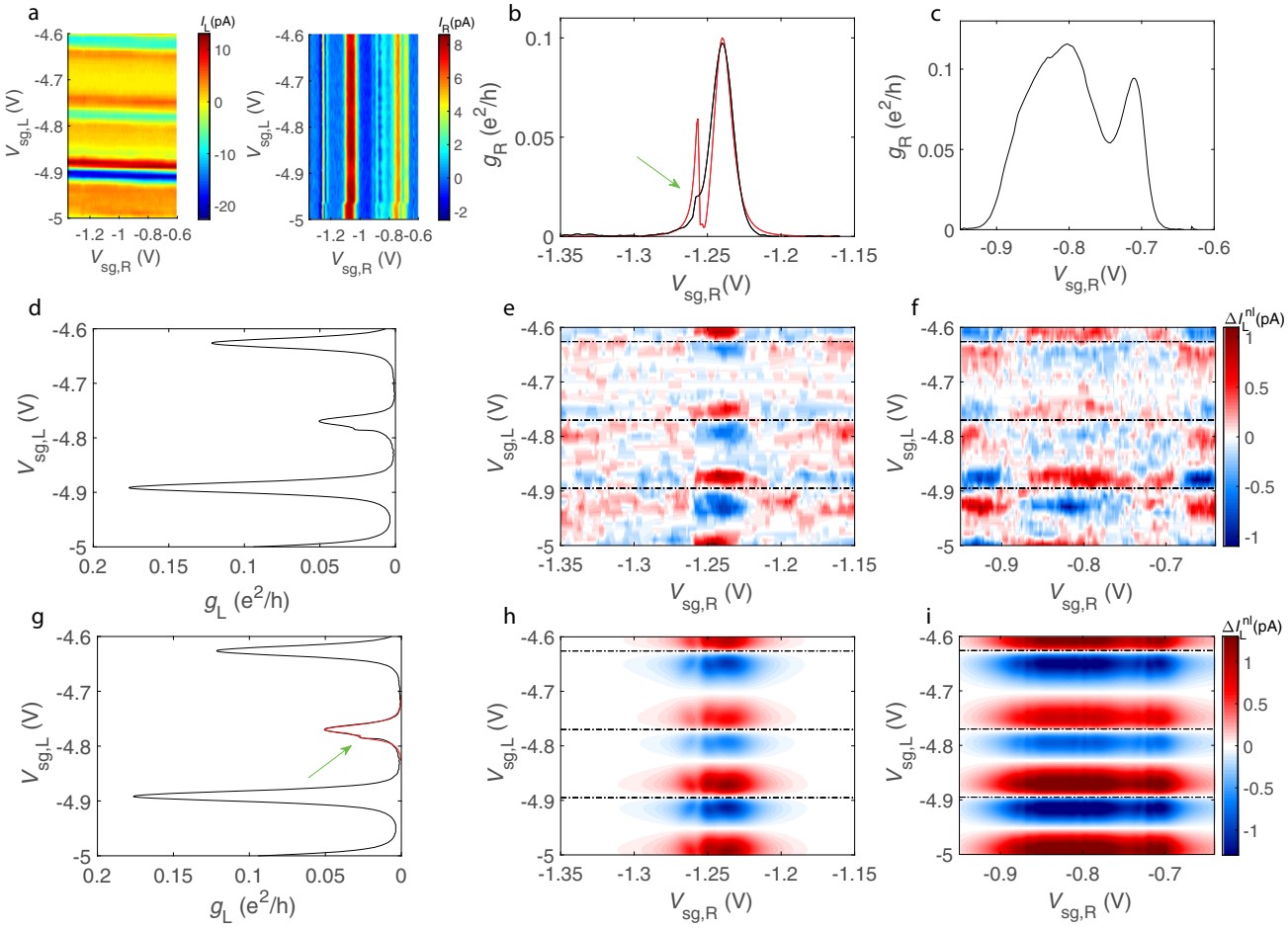

**Fig. 2 Local and non-local contributions to the thermoelectric current. a** Thermally generated current at $2f$ in the left and right dot measured as a function of $V_{sg,L}$ and $V_{sg,R}$; the data include both local and non-local thermoelectric contributions. **b**, **c** Zero bias conductance of the right quantum dot vs. $V_{sg,R}$ in two intervals: $-1.35\,V < V_{sg,R} < -1.15\,V$, and $-0.95\,V < V_{sg,R} < -0.6\,V$; green arrow in the **b** points to the minor peak in the vicinity of the main conductance peak, and the red curve is the fit by the Fano resonance model with the parameters $\Gamma_R = 20\,\mu eV$, $\gamma_R = 252\,\mu eV$, $\varepsilon_R - \varepsilon'_R = 120\,\mu eV$, $t_R = 55\,\mu eV$; the additional Fano peak in the fit is intentionally made stronger than the experimentally observed one in order to better reproduce the behavior of the non-local thermoelectric current in Fig. 3d. **d** Zero bias conductance of the left quantum dot in the interval $-5\,V < V_{sg,L} < -4.6\,V$. **e**, **f** Experimental non-local contribution to the thermal current in the left quantum dot, $\Delta I_L^{nl}$, in two selected regions of $(V_{sg,R}, V_{sg,L})$ plane. **g** Zero bias conductance of the left quantum dot; data as in **d**, but the red curve displays the fit of one of the peaks with the Fano resonance model with the parameters $\Gamma_L = 6\,\mu eV$, $\gamma_L = 98\,\mu eV$, $\varepsilon_L - \varepsilon'_L = 24\,\mu eV$, and $t_L = 10\,\mu eV$. **h**, **i** Theoretically predicted non-local contribution $\Delta I_L^{nl}$ in the same regions as in **e**, **f**.

The thermoelectric current induced by the heater in the left (right) quantum dot is given by the sum of dominating local ($I_{L(R)}^{loc}$) and small non-local contributions ($\Delta I_{L(R)}^{nl}$), $I_{L(R)} = I_{L(R)}^{loc}(V_{sg,L(R)}) + \Delta I_{L(R)}^{nl}(V_{sg,L}, V_{sg,R})$, see Supplementary Note 4. To infer the non-local contribution from the measured current $I_{L(R)}$, we subtract off its slowly varying average local background, $\langle I_{L(R)} \rangle$ (see "Methods"):

$$\Delta I_{L(R)}^{nl} = I_{L(R)} - \langle I_{L(R)} \rangle. \qquad (3)$$

We thus obtain non-local currents $\Delta I_{L(R)}^{nl}$, which have a magnitude of order of 5–10% of the total thermoelectric currents. Figure 2 displays the maps of the non-local thermoelectric current in left dot $\Delta I_L^{nl}(V_{sg,L}, V_{sg,R})$ measured in the vicinity of the two conductance peaks of the right dot for the heating voltage $V_h = 5\,mV$. In Fig. 2b–d, g, $g_{L(R)} = hG_{L(R)}/e^2$ is the dimensionless conductance of the left (right) quantum dot. We find that $\Delta I_L^{nl}$ is symmetric with respect to the centers of the conductance peaks of the right dot, but it changes sign at the maxima of conductance peaks of the left dot. Thus the non-local

current $\Delta I_L^{nl}$ approximately follows the same pattern as the product $[dg_L(V_{sg,L})/dV_{sg,L}]g_R(V_{sg,R})$.

Before proceeding to our main result, note that some conductance peaks are split into two closely located peaks (see Fig. 2b, c). The splitting is explained by the Fano resonant effect, see Supplementary Note 4. Namely, we introduce the coupling rates $\Gamma_{j,n}$ and $\gamma_{j,n}$ (here $j = L, R$ enumerates the dots) between the $n$th energy level of the dot (with energy $\varepsilon_{j,n}$) and, respectively, normal and superconducting leads; we also assume that the $n$th level is coupled to a dark energy level, having the energy $\varepsilon'_{j,n}$, via the hopping matrix element $t_{j,n}$. This results in the transmission probabilities of the dots $\tau_j = \sum_n \gamma_{j,n}\Gamma_{j,n}/[[(E - \varepsilon_{j,n} - |t_{j,n}|^2/(E - \varepsilon'_{j,n}))^2 + (\gamma_{j,n} + \Gamma_{j,n})^2/4]$, see Supplementary Note 4. The conductances $g_L(V_{sg,j}) = 4\tau_j^2(0, V_{sg,j})/[2 - \tau_j(0, V_{sg,j})]^2$, as predicted by the theory of Andreev reflection[43], exhibit splitted peaks for $t_{j,n} \neq 0$. In our model, we ignore Coulomb interaction because the charging energies of the dots are relatively small, $E_{C,j} \lesssim \gamma_{j,n} + \Gamma_{j,n}$.

**Non-local thermoelectricity.** Figure 3 displays the main result of our study. There we plot the non-local thermal currents for both

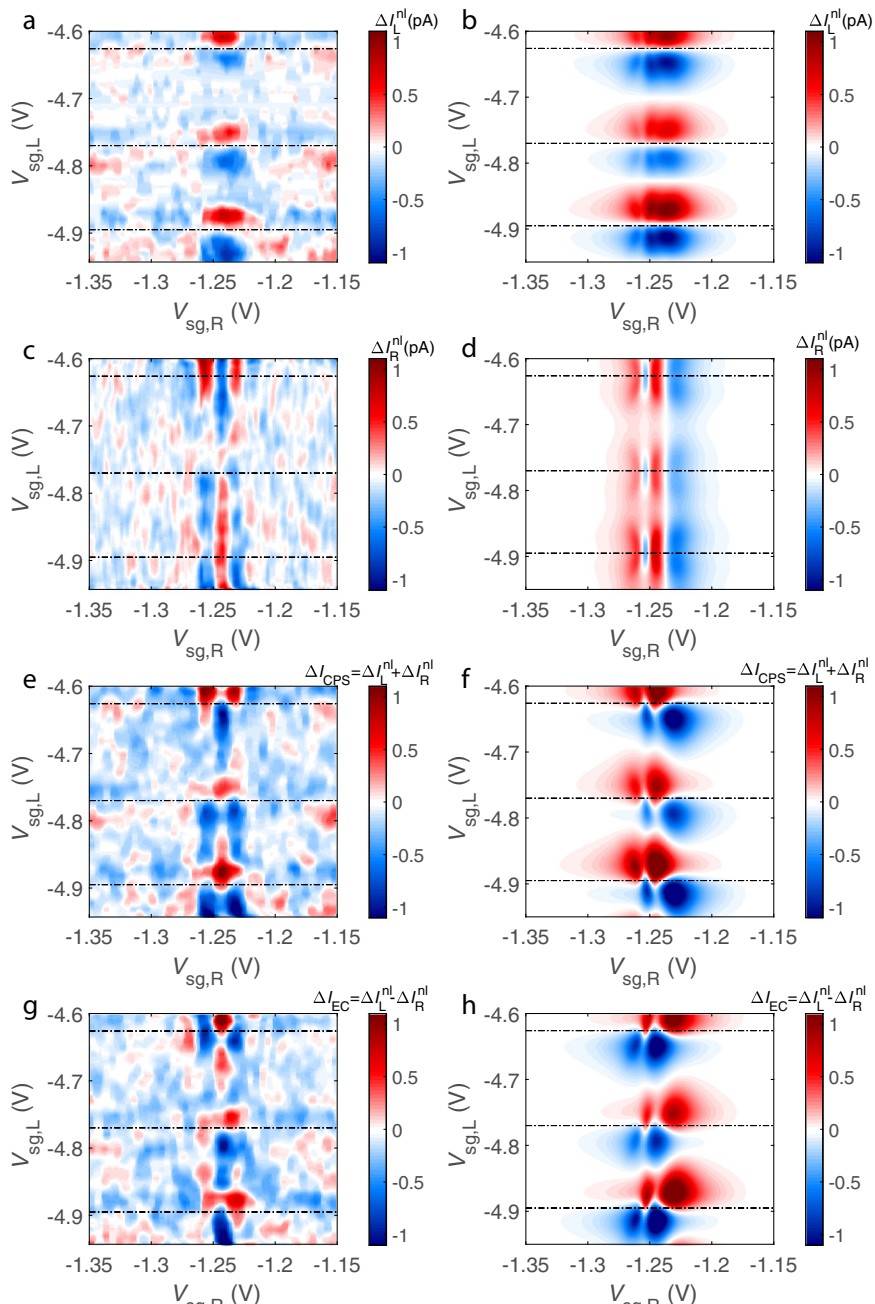

**Fig. 3 Interplay of non-local thermoelectric currents in the left and right dots.** Non-local contributions to the thermal currents of the right and left dots (**a–d**) and the corresponding CPS and EC currents (**e–h**). The left column (graphs **a**, **c**, **e**, **g**) shows the experiment and the right one (graphs **b**, **d**, **f**, **h**)—theory. Horizontal dotted lines in the plots indicate the positions of the conductance peak maxima of the left dot.

quantum dots together with the theory predictions based on Eqs. (1) and (2). The involved model parameters are chosen in such a way that, besides accounting well for the non-local current, they can also reasonably fit the conductance peaks (see the caption of Fig. 2). In the experiment, the non-local current $\Delta I_R^{nl}$ changes its sign three times in the vicinity of the conductance peak of the right dot located at $V_{sg,R} = -1.24$ V. Although in order to reproduce this behavior we had to take hopping amplitude, $t_R$, larger than required by the perfect fit to the conductance peak (see Fig. 2b), this offers a fair cross-check for our description. One sees that not only the magnitudes of the currents $\Delta I_L^{nl}$ and $\Delta I_R^{nl}$ are in good agreement with the theory, but their symmetric and anti-symmetric combinations $\Delta I_{CPS} = \Delta I_L^{nl} + \Delta I_R^{nl}$ and $\Delta I_{EC} = \Delta I_L^{nl} - \Delta I_R^{nl}$ exhibit the expected

gate voltage dependence, although the comparison for $I_{EC}$ is hampered by large noise as the EC current is a difference between two small non-local signals. The observed general agreement in Fig. 3 provides strong support of the non-local coherent thermoelectric effect in our device.

**Local thermoelectricity.** Since, as noted, the non-local currents are relatively small, one can treat the measured currents foremost as local, $I_{L(R)} \simeq I_{L(R)}^{loc}$. The measured thermoelectric current of the left quantum dot is shown in Fig. 4a. The lowest curve in this panel shows the dimensionless conductance of the left quantum dot $g_L = hG_L/e^2$ as a function of the side gate voltage $V_{sg,L}$. Thermoelectric current $I_L$, depicted by the upper curves of Fig. 4a, varies with the

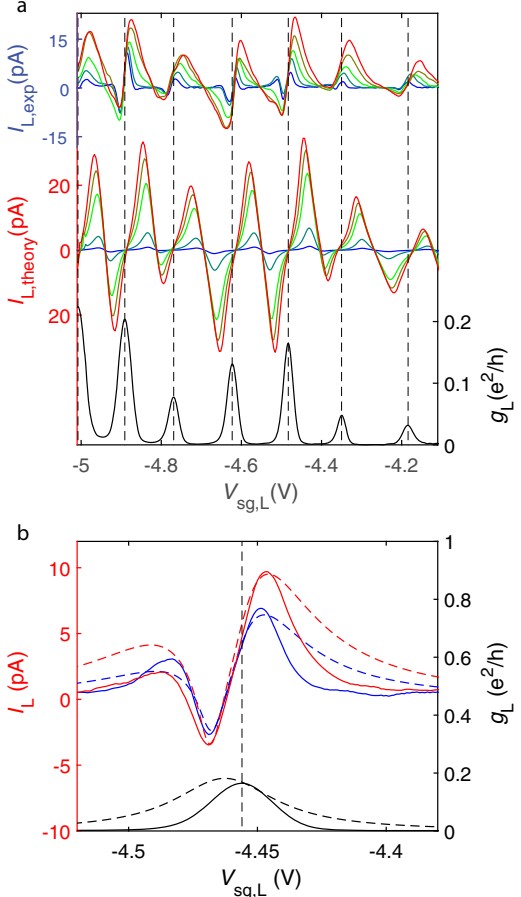

**Fig. 4 Drive dependence of thermoelectric current. a** Upper curves: thermoelectric current in the left quantum dot $I_{L,exp}$ vs. gate voltage $V_{sg,L}$ measured at heating voltages $V_h = [5, 9, 19, 25, 29]$ mV, where $V_h = 5$ mV is the blue curve and 29 mV is the red curve. We estimate the induced temperature difference between the left and right quantum dots to be $T_L - T_R \simeq 17$ mK for $V_h = 5$ mV, and $T_L - T_R \simeq 59$ mK for $V_h = 29$ mV. Middle curves: theory predictions based on the coherent model (see Supplementary Note 4) for the thermoelectric current $I_{L,theory}$ plotted in the same manner as the upper curves for $V_h = [5, 10, 20, 25, 30]$ mV. The gap of Al/Ti leads is set to $\Delta_0 = 150$ μeV at $T = 0$ while the BCS gap formula $\Delta$ $(T_S)$ with $T_S = (T_L + T_R)/2$ defines the $T$ dependence. Lowest curve: experimental conductance of the left quantum dot. **b** Incoherent modeling for the low temperature regime: theoretical fits (dashed) to the measured thermoelectric currents $I_L$ and conductance $g_L$ (solid) at $V_h = 7$ and 11 mV, blue and red curves, respectively. The model is based on the assumption that the system may be split into the coherent subsystems, which, in turn, are joined incoherently into a circuit. Details of the model and fitting parameters are given in Supplementary Note 5.

same period as the conductance. Its magnitude grows with the increasing heating power $P$ as $I_L^{max} \propto P^{1/3}$, which is consistent with $G_{th} \propto T^3$ for the thermal conductance between electrons in graphene and phonons in the substrate. The maximum thermal power of the left quantum dot reaches a value of $\alpha_{max} = \max\{I_L/G_L(T_L - T_S)\} \approx 250$ μV/K, which is close to the values reported in ref. [42]. Since we cannot reliably measure the temperature of the superconductor $T_S$, we set $T_S = (T_L + T_R)/2$ in evaluating $\alpha_{max}$ and in our theory modeling. In Fig. 4a, we also show the local thermoelectric current predicted by the theory of Andreev reflection with energy-dependent transmission probability[44]; the same theory was earlier employed in deriving the non-local contributions using Eq. (2). In the local case, only those quasiparticles

with energies above the superconducting gap, $|E| > \Delta$, contribute. The zero temperature value of the gap $\Delta_0$ is set by the transition temperature $T_c = 1.0$ K of the Al/Ti leads, and the transmission probability of the dot $\tau_L(E, V_{sg,L})$ is inferred from the experimentally measured conductance $g_L(V_{sg,L})$, as explained in Supplementary Note 5. We find rather good agreement between theory and experiment except for the lowest values of the heating voltage. This agreement provides further confirmation for our model.

In the low temperature regime, the coherent model predicts very small current due to lack of quasiparticles, while the experimental thermoelectric current remains significant and exhibits additional sign changes in the vicinity of some of the conductance peaks. These features can originate from non-zero, thermally induced voltages across the dots. To capture these effects, we propose that electrons may undergo quick inelastic relaxation, see Supplementary Note 5. This introduces incoherent effects that facilitate description of quantum dots and NS interfaces as independent conductor elements connected in series. The results of such an inelastic model are shown in Fig. 4b. The incoherent description accurately predicts the character of the local thermoelectricity at small temperatures. Incidentally, although at odds with the effect of local thermoelectricity, the non-local currents are dominantly determined by coherent electrical transport.

Alternatively, the additional peak in the local thermoelectricity could originate from Coulomb blockade[3,5] as the non-local thermoelectric effect is shown to develop from single peak to double peak structure when temperature is lowered[5]. The calculated double peak structure is similar to our local thermoelectric signal observed at low temperatures. However, the resonance energy dependence of the non-local signal with Coulomb blockade agrees poorly with the experimental results in Fig. 3a in comparison with our coherent transport model.

## Discussion

This work has demonstrated the use of thermal gradient as primus motor for generating entangled electrons in graphene Cooper pair splitter. As the quantum dots in the device can be tuned individually, we are able to tune the device operation between EC and CPS regimes, thereby accomplishing direct control of two streams of entangled electrons. This type of scheme is useful not only for enabling devices where electrical drive is neither possible nor desired but also as a platform for realizing quantum thermodynamical experiments.

## Methods

**Samples and fabrication.** Our graphene films were manufactured using mechanical exfoliation of graphite (Graphenium, NGS Naturgraphit GmbH) and placed on a highly p[++] doped silicon wafer, coated by 280-nm-thick thermal silicon dioxide. The conducting substrate was employed as a backgate for coarse tuning of the graphene quantum dots, while fine tuning was performed by adjusting the side gates. Electron beam lithography (EBL) on PMMA resist was used to pattern a mask for plasma etching of the graphene structures. A second EBL step was carried out to expose the pattern for electrode structures, followed by deposition of Ti/Al (5/50 nm, superconducting $T_c = 1.0$ K) leads using an e-beam evaporator. Normal contacts to the graphene quantum dots were made using etched graphene nanoribbons with a small number of conductance channels at the operating point[45].

The strong p[++] doping and the interfacial scattering at the Si/SiO₂ interface reduce the phonon mean free path in the substrate to one micron range, which facilitates the use of the heat diffusion equation for estimating thermal gradients along the substrate near the graphene ribbon heater and the splitter. Heat transport analysis was done separately for each component involved in the operation of the CPS, as well as a COMSOL simulation, see Supplementary Note 1.

**Measurement scheme.** Our conductance and thermoelectric current measurements employed regular lock-in techniques at low frequencies. In the thermoelectric experiments, we had one DL1211 current preamplifier connected to each quantum dot, while the superconductor was grounded on top of the cryostat. The current gain of DL1211 amplifier was set to $10^6$ V/A, which provides a virtual ground of 20 Ω. The lead resistance including filters was approximately 100 Ω, which is much less than the quantum dot resistance ~$h/e^2$, the quantum resistance. The galvanically separated

heater was driven at $f = 2.1$ Hz, with an ac voltage amplitude ranging between 1 and 40 mV (for data without galvanic separation, see Supplementary Note 3). Because the resistance $R$ of the graphene ribbon heater was independent of temperature in its regime of operation, the heating power $P = V_h^2/R$ was fully governed by the voltage $V_h$. The heating power oscillated at frequency $2f = 4.2$ Hz, which resulted in thermoelectric currents at 4.2 Hz, recorded using a lock-in time constant of 1 s. The thermal response time of our device appears to be well below 1 ms, i.e., much less than a measurement period, so that the thermal response is not suppressed. The use of such a low frequency for the experiments was dictated by the need to eliminate the capacitive coupling between the wires in the measurements.

The local temperature was monitored using two SGS junctions. At low temperature, because of the proximity effect, graphene becomes superconducting, with a supercurrent exponentially proportional to temperature: $\sim \exp(-T/E_{Th})$. Here $E_{Th} = \hbar D/L_{SGS}^2$ stands for the Thouless energy given by the length of the SGS section $L_{SGS}$ and the diffusion constant $D \sim \frac{1}{2} v_F \lambda$, where the Fermi velocity $v_F = 8 \times 10^5$ m/s and $\lambda$ is the charge carrier mean free path of graphene. Using $\lambda \simeq 20$ nm for graphene on SiO$_2$ and $L_{SGS} = 200$ nm, we estimate $E_{Th} \simeq 130$ μeV for our SGS junctions. This kind of SGS junctions in the intermediate length regime ($E_{Th} \simeq \Delta$) were experimentally found to provide good thermometers over the relevant range of temperatures in our work.

We have used the amplitude of the differential resistance peak $R_d^{max}$ vs. $T$ to infer the effective local temperature within the graphene sample. The SGS temperature under the voltage bias $V_h$ in the graphene ribbon heater was obtained by direct comparison between $R_d^{max}$ and the heating power to the value of $R_d^{max}$ recorded when varying the cryostat temperature. As detailed in Supplementary Note 2, we obtain the relation $T_{SGS,L} = 9.1 \times V_h^{0.70} + 90$ mK between the SGS$_L$ temperature and the heating voltage ($V_h$ in Volts). For the SGS$_R$ thermometer, we obtained an estimate $T_{SGS,R} \simeq 8.4 \times V_h^{0.70} + 90$ mK.

**Background subtraction**. The direct experimental measurement allows to obtain the total electric current through the left (right) dot, $I_{L(R)}$, constituting the sum of local, $I_{L(R)}^{loc}$, and non-local, $\Delta I_{L(R)}^{nl}$, contributions. Note that, in theory, while the non-local contribution depends on both gate voltages, the local one is determined only by the gate voltage on the corresponding dot. This suggests that the local current through the left (right) dot $I_{L(R)}^{loc}$ is nothing but the total current $I_{L(R)}$ averaged over the gate voltage on the opposite dot; $\Delta I_{L(R)}^{nl}$ can be thus obtained by subtracting off this average background from $I_{L(R)}$. On practice, however, the gate electrodes may be subject to cross-talk, which the described simple processing does not account for. For the most part, the cross-talk was eliminated by a small rotation of the data array. Bearing in mind the remaining residual cross-talk, we construct a slowly varying background $\langle I_{L(R)}(V_{sg,L}, V_{sg,R}) \rangle$ in the following manner (for clarity, let us consider the case of the left dot): for any fixed $V_{sg,L} = V_{sg,L}^F$, $\langle I_L(V_{sg,L}^F, V_{sg,R}) \rangle$ is the linear fit of $I_L(V_{sg,L}^F, V_{sg,R})$ as function of $V_{sg,R}$. The non-local contribution is then obtained using the formula

$$\Delta I_L^{nl}\left(V_{sg,L}, V_{sg,R}\right) = I_L\left(V_{sg,L}, V_{sg,R}\right) - \left\langle I_L\left(V_{sg,L}, V_{sg,R}\right)\right\rangle. \quad (4)$$

Linear fits in the background construction were found sufficient for compensating the remaining residual tilt of the current maps on the $\{V_{sg,L}, V_{sg,R}\}$ plane.

**Theoretical modeling**. Our theoretical calculations are based on both coherent and incoherent modeling of transport. In coherent modeling, we employ the Landauer approach with Andreev reflection[43,46] for calculating the local thermoelectric current; Lorentzian resonance line shapes are employed for transport in the quantum dots[35,44,47,48]. For the non-local current, we employ a standard crossed Andreev reflection formalism[2,49–51]. In our incoherent theory, based on scattering matrix formalism[1,4,52], we also include the influence of internal thermally generated current sources and their back-action effect owing to the environmental impedance caused by graphene ribbons. The inclusion of the back-action-induced voltage sources makes the incoherent calculation self-consistent. For details of the calculations, we refer to Supplementary Notes 4 and 5.

## Data availability

All data needed to evaluate the conclusions in the paper are covered by the paper and its Supplementary Information. Additional data related to this work are available from authors upon reasonable request.

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

## Acknowledgements

We are grateful to C. Flindt, P. Burset, and T. Heikkilä for discussions and to I. A. Sadovskyy for sharing his numerical codes. This work was supported by Aalto University School of Science Visiting Professor grant to G.B.L., as well as by Academy of Finland Projects No. 290346 (Z.B.T., AF post doc), No. 314448 (BOLOSE), and No. 312295 (CoE, Quantum Technology Finland). The work of A.L. was support by the Vilho, Yrjö and Kalle Väisälä Foundation of the Finnish Academy of Science and Letters. This work was also supported within the EU Horizon 2020 program by ERC (QuDeT, No. 670743), and in part by Marie-Curie training network project (OMT, No. 722923), COST Action CA16218 (NANOCOHYBRI), and the European Microkelvin Platform (EMP, No. 824109). The work of N.S.K. and G.B.L. was supported by the Government of the Russian Federation (Agreement No. 05.Y09.21.0018), by the RFBR Grants No. 17-02-00396A and No. 18-02-00642A, Foundation for the Advancement of Theoretical Physics and Mathematics BASIS, the Ministry of Education and Science of the Russian Federation No. 16.7162.2017/8.9. The work of N.S.K and A.G. at the University of Chicago was supported by the NSF grant DMR-1809188. The work of V.M.V. was supported by the U.S. Department of Energy, Office of Science, Basic Energy Sciences, Materials Sciences and Engineering Division.

## Author contributions

This research, initiated by P.J.H., is an outgrowth of a long-term collaboration between G.B.L. and P.J.H. The experimental setting and the employed sample configuration were developed by Z.B.T. and P.J.H. The patterned graphene samples were manufactured by Z.B.T. using Aalto University OtaNano infrastructure. The experiments were carried out at OtaNano LTL infrastructure by Z.B.T. and A.L. who were also responsible for the data analysis. A.S. and M.H. were adjusting and operating the LTL infrastructure. Theory modeling for coherent transport was performed by D.S.G., N.S.K., and G.B.L. The theory for incoherent transport was foremost developed and analyzed by N.S.K., A.G., V.M.V., and G.B.L. The results and their interpretation were discussed among all the authors. The paper and its Supplementary Information were written by the authors together.

## Competing interests

The authors declare no competing interests.
