## [Peer Review File · Nature Communications]

Reviewer #1 (Remarks to the Author):

The paper presents a very interesting experiment on the measurement of the first (as far as I know) nonlocal thermoelectricity in a graphene-based double-dot Cooper pair splitter.

A very timely argument since different theoretical papers Ref3-5 in the last year have considered such phenomenology. The experiment is really challenging since there is the need to control thermal gradients at the submicrometer scale at sub-kelvin temperatures making a multiterminal measurement (in order to test nonlocality). Also, the data analysis and the comparison with the theoretical model is quite elegant and convincing.

My opinion on the experiment is very positive in general.

It is an excellent experimental work.

I have only a few main reservations which I ask the author to reply and eventually clarify the paper on these points

1) I have some problems to accept that for low-temperature regime an incoherent regime becomes more appropriate. This may be the consequence of the fact that the experimental procedure missed to keep properly shorted circuited the three terminals as should be done in order to forbid extra unwanted currents contributions.

In superconducting circuits, the current can be measured very precisely without the need to localized a d.d.p. at the measurement terminals.

The authors did not clearly explain how the current is indeed measured. Is indeed measured with an external ammeter or what else? In particular, I wish to know how the effective resistance of the current measurement (if any) compare with the intrinsic resistance of the graphene terminals.

I am also thinking if the incoherent regime would be less relevant if the short-circuit condition is fully imposed. Further, the authors should provide an explanation of why this happens at low temperatures. I would expect that incoherent behavior is justified at higher temperatures and not at low. What I am missing?

2) The authors seem to completely rule out Coulomb effects. But the dimensionless conductance is not so high at least looking Fig.2b-2c-2d-2g. I agree that probably this does not change too much for the validity of the nonlocal measurement presented in Fig2 and Fig3, but, eventually, it may be one reason behind the unexpected "incoherent" regime. I have seen some signs in the conductance peak which may be even connected to Coulomb interaction, such as the small "near resonance peaks" or the small shoulder in the resonances. In the supplementary, they discuss those assumptions but, in the main text, they do not comment on it.

Note that theory of nonlocal thermoelectricity in double dot with Coulomb interaction (such as the theory presented in Ref. 5) shows that when the temperature is small $\Gamma_S \gg k_B T$ (with $T = (T_L + T_R)/2$) and the Andreev bound states may be resolved and two thermoelectrical resonant peaks are observed. At higher temperatures $\Gamma_S \sim T$ the thermoelectrical peaks merge to a single thermoelectrical resonant peak.

Is it possible that the difference between the non-local thermoelectricity at low and high temperatures could be explained in these terms?

Could the author see (if they have the data) what happens to a specific non-local thermoelectric resonance in terms of the difference $\epsilon_R - \epsilon_L$ at different temperatures T ? Because of the disappearance of a sign change sound similar to this phenomenology.

3) The small contribution of the nonlocal component with respect to the local one is mainly given by the fact that the gap of the superconductor (Al film) is too near to the operating temperature. Increasing gap, the contribution of local and nonlocal EC contributions should become less important with respect to CPS contribution. Is this correct? Is this progression eventually observed in some of the data taken in the experiment? Could a different superconductor such as Nb be considered?

4) Honestly, excluding Fig 3a and Fig 3b, the comparison between theory and experiment is not very satisfactory weak, especially for d and h. Could the author try to find the possible reasons and comment on it better?

After these clarifications I can see if their statements are sustained from the data presented. I am aware that it is a complex experiment so I do not ask extra measurement but only other data analysis if they have it.

Let's come to minor points on the paper

a) On the bibliography I feel fair to ask a more complete reference to recent literature in particular: concerning the thermoelectricity in superconducting and multiterminal systems probably it is fair to mention also this recent literature

"Thermoelectric properties of microstructures with four-probe versus two-probe setups" Glen D. Guttman, Eshel Ben-Jacob, and David J. Bergman Phys. Rev. B 53, 15856 (1996)

"Thermoelectric efficiency of three-terminal quantum thermal machines" Francesco Mazza et al New J. Phys. 16 085001 (2014)

"Resolving thermoelectric "paradox" in superconductors" Connor D. Shelly et al. Science Advances 2, e1501250 (2016)

"Local and nonlocal thermopower in three-terminal nanostructures" G. Michałek, et al. Phys. Rev. B 93, 235440 (2016)

"Thermopower and thermal conductance of a superconducting quantum point contact" Sergey S. Pershoguba and Leonid I. Glazman Phys. Rev. B 99, 134514 (2019)

"Nonlinear Thermoelectricity with Electron-Hole Symmetric Systems" G. Marchegiani et al. Phys. Rev. Lett. 124, 106801 (2020)

"Nonlocal Thermoelectricity in a Superconductor–Topological-Insulator–Superconductor Junction in Contact with a Normal-Metal Probe: Evidence for Helical Edge States" Gianmichele Blasi et al. Phys. Rev. Lett. 124, 227701(2020)

b) Comment on the role of Coulomb interaction in the introduction and in some part of the main text

c) Tell somewhere (probably in the measurement scheme how the current is measured, think the point the 1 questioned before). Maybe the electrical measurement scheme can be sketched in Fig.1 If the superconducting lead is kept at the same voltage of the graphene terminals tell it in the main text specifically (maybe around Eq.2).

d) In the main text tell what is the substrate at pag4. Even if this is specified in the method and Supplementary a clarification in main text it is useful.

e) Define better the meaning of the average in Eq.3

f) In the supplementary frequently the name thermal current is used in place of the thermoelectrical current. In order to not create confusion could you check if the “thermal” before Eq12 and at the end of pag12 is correct. Thermal current is usually used to express the heat current but here I assume want to express the charge current generated by the difference of temperature (i.e thermoelectricity local or nonlocal depending on the situation). Please check also the main text

Once all these observations are satisfactory answered and replied I am ready to consider supporting the publication on Nature Comm.

Best regards

Reviewer #2 (Remarks to the Author):

Re:

Code: NCOMMS-20-18175-T

Authors: Zhenbing Tan, Antti Laitinen, Nikita Kirsanov, Alexey Galda, Mohammad Haque, Alexander Savin, Dmitry Golubev, Valerii Vinokur, Gordey Lesovik, and Pertti Hakonen

Title: Thermoelectric current in a graphene Cooper pair splitter

In the present manuscript, the authors present measurements and theoretical calculations of thermoelectric currents in a so-called Cooper pair splitter. The Cooper pair splitter is fabricated in a nanostructured device containing graphene and superconducting electrodes. The basic effect is that a superconducting injector distributes the two electrons of a Cooper pair into two, spatially separated normal metal terminals. Such a process can happen due to a purely thermal bias at zero voltage difference, as was theoretically predicted several times earlier. The authors report the measurement of the electric current due to selective heating of one of the normal sides and can control these currents using electrostatic gates. Analyzing the data the authors extract the Seebeck coefficient, and they find for the nonlocal response values as high as $250 \mu\text{V}/\text{K}$. The authors relate their findings to the creation of entangled electron pairs due to the Cooper pair splitting.

The overall impression of the manuscript is positive. The experimental results are interesting and novel and bear perspectives for the field of quantum thermodynamics. On one side, the fabricated device is very sophisticated and the measurement seems solid. The experimental signature that a finite nonlocal electric current is induced by just a temperature gradient is very clearly visible. On the other side, there are several issues in which the manuscript is not convincing. Many of those are related to the theoretical explanations, which the authors need to address before the manuscript should be considered further.

- In the very beginning and throughout the introduction, the authors emphasize the relation of Cooper pair splitting to entanglement. Now, the experiments performed for the manuscript do not allow such a conclusion since they don't test the actual entanglement of the split streams of electrons. Hence, I would recommend formulating here more carefully. Furthermore, as the authors show, the results are described by an incoherent model that neglects any quantum entanglement. One could, therefore, conclude the authors have shown that no entanglement has been produced in

the present device.

- While the authors discuss and cite many works on the measurement of the thermopower in quantum dots, they miss in my view several important experimental works on the thermoelectric effect in superconducting heterostructures. E.g. in Kolenda et al., PRL 116, 097001 (2016) a thermopower of similar size was found in a superconductor-ferromagnet junction. A recent review of the field is found in A. Fornieri and F. Giazotto, Nat. Nanotechnol. 12, 944 (2017).

- Cooper pair splitting by a thermal gradient was also studied in detail in Z. Cao, T.-F. Fang, L. Li, and H.-G. Luo, Appl Phys Lett 107, 212601 (2015). Even the model used in that paper seems to be quite similar to the one used in the present manuscript.

- About the model, the authors claim to use scattering theory, but on the other hand mention incoherent transport. This is somewhat contradictory. Also, a formula like $\tau_L(E) \cdot \tau_S \cdot \tau_R(-E)$ seems to represent just the product of three probabilities. How does this fit to the scattering approach? Furthermore, τ_S is calculated from the Usadel theory. Again this is not obviously compatible with the scattering approach. Note that an incoherent series addition of three elements would just add resistances, which are proportional to the inverse of the 'probabilities'.

- For the determination of the Seebeck coefficient, a precise measurement of the temperature differences is necessary. However, I have the impression, the SGS thermometers are some distance away from the quantum dots. So, how exactly was the temperature calibrated? In addition, there is an issue with the determination via the SGS supercurrent, see below.

- When describing the measurement of the local temperature, the authors use the temperature dependence of the supercurrent in the SGS structure. However, contrary to the authors' claim, the correct formula is different from the ones used in the manuscript. Namely, the dependence is rather of the form $\sim \exp(-a \cdot \sqrt{T/T^*})$ in the diffusive limit (see e.g. J. Low Temp. Phys. 106, 305 (1997) for a detailed discussion). How does this affect the measurement scheme? Furthermore, the authors calculate $E_{Th} \sim 1305 \mu\text{eV}$, which according to my estimate rather corresponds to a temperature of 13 K. If that is correct the junctions operate at a temperature much smaller than the Thouless energy and the supercurrent is basically temperature independent. Finally, what defines the 'optimum thermometer sensitivity'?

- In Figure 4 a direct comparison between theory and experiment is presented. A strange feature in panel a is the "symmetry" of the current peaks in the experiment that seems to be opposite to the theory. In experiment, the peaks seem to 'lean' towards the dashed line, whereas in theory they 'lean' away. This is not an essential issue, but maybe the authors have an idea, where this discrepancy comes from.

REVIEWER COMMENTS

Reviewer #1 (Remarks to the Author):

The paper presents a very interesting experiment on the measurement of the first (as far as I know) nonlocal thermoelectricity in a graphene-based double-dot Cooper pair splitter.

A very timely argument since different theoretical papers Ref3-5 in the last year have considered such phenomenology. The experiment is really challenging since there is the need to control thermal gradients at the submicrometer scale at sub-kelvin temperatures making a multiterminal measurement (in order to test nonlocality). Also, the data analysis and the comparison with the theoretical model is quite elegant and convincing.

My opinion on the experiment is very positive in general.
It is an excellent experimental work.

I have only a few main reservations which I ask the author to reply and eventually clarify the paper on these points

1) I have some problems to accept that for low-temperature regime an incoherent regime becomes more appropriate. This may be the consequence of the fact that the experimental procedure missed to keep properly shorted circuited the three terminals as should be done in order to forbid extra unwanted currents contributions.

In superconducting circuits, the current can be measured very precisely without the need to localized a d.d.p. at the measurement terminals.

The authors did not clearly explain how the current is indeed measured. Is indeed measured with an external ammeter or what else? In particular, I wish to know how the effective resistance of the current measurement (if any) compare with the intrinsic resistance of the graphene terminals.

I am also thinking if the incoherent regime would be less relevant if the short-circuit condition is fully imposed. Further, the authors should provide an explanation of why this happens at low temperatures. I would expect that incoherent behavior is justified at higher temperatures and not at low. What I am missing?

Our response (all changes are in bold in the attached manuscript version):

We thank the Reviewer for pointing out the insufficiency of relevant details in the experimental setting. For our measurements, a current preamplifier DL1211 was connected to amplify the thermoelectrical current. The current amplifier was operated at the transresistance gain of 10^6 V/A which provides a virtual ground with 20 Ohms that is much smaller than the resistance of the quantum dots. The measurement lead resistance including filters was ~100 Ohms, which is also fine considering the effective short-circuiting conditions. We added these details into the Methods section.

As discussed a bit more later, we do not fully understand on the microscopic level the local transport properties of our structure at the lowest temperatures at which Coulomb interaction, Kondo effect and possibly other non-trivial effects may become important. Our incoherent transport model should be viewed as a phenomenological way of taking these effects into account.

2) The authors seem to completely rule out Coulomb effects. But the dimensionless conductance is not so high at least looking Fig.2b-2c-2d-2g. I agree that probably this does not change too much for the validity of the nonlocal measurement presented in Fig2 and Fig3, but, eventually, it may be one reason behind the unexpected "incoherent" regime. I have seen some signs in the conductance peak which may be even connected to Coulomb interaction, such as the small "near resonance peaks" or the small shoulder in the resonances. In the supplementary, they discuss those assumptions but, in the main text, they do not comment on it.

Note that theory of nonlocal thermoelectricity in double dot with Coulomb interaction (such as the theory presented in Ref. 5) shows that when the temperature is small $\Gamma_S \gg k_B T$ (with $T = (T_L + T_R)/2$) and the Andreev bound states may be resolved and two thermoelectrical resonant peaks are observed. At higher temperatures $\Gamma_S \sim T$ the thermoelectrical peaks merge to a single thermoelectrical resonant peak.

Is it possible that the difference between the non-local thermoelectricity at low and high temperatures could be explained in these terms?

Could the author see (if they have the data) what happens to a specific non-local thermoelectric resonance in terms of the difference $\epsilon_R - \epsilon_L$ at different temperatures T ? Because of the disappearance of a sign change sound similar to this phenomenology.

Our response:

Indeed, we neglected the Coulomb blockade effects in the theoretical model. We did that for several reasons. First of all, our estimates yield the charging energy of the left (right) quantum dot to be around 100 (250) μeV , which is close to the coupling energy between the dot and the lead γ_L (γ_R). It is known from the theory that if $E_C < \gamma_{L,R}$, the Coulomb blockade is significantly weakened. We should mention, however, that it is difficult to extract reliable values of charging energies from the experimental data because E_C is much smaller than the level spacing in the dots. Second, the I-V curves of the dots do not exhibit a sharp Coulomb gap even at low temperatures, which also points out to the weakness of the Coulomb blockade effects. Third, the non-local signal, which is our main focus, is not very sensitive to the nature of the local transport through the dots. Namely, Eq. (1) should be valid even in the presence of the Coulomb blockade in the quantum dots. The charging effects in the dots are then incorporated into the effective transmissions $\tau_{L,R}(E)$, which can be extracted from measured zero bias conductances of the dots. Finally, last but not least, given that our model nicely fits the data at higher temperatures, we preferred to refrain from the sophistications which, although capable to capture subtle low-

temperature features observed at the lowest temperatures, would turn our model excessively complex and less transparent.

At the same time, we certainly agree with the Reviewer that Coulomb blockade may play role at the lowest temperatures explored. Moreover, we would indeed expect that these effects may be responsible for splitting of some of the conductance peaks. To take this into account, we compared our data with the predictions of the theory of Ref. [5]. However, the theory did not produce the right shape of the double-peak structure in the local thermal current shown in Fig. 4b. Several other theoretical models have been employed but did not offer much success. Given the overall situation, it was found most productive for the successful description of the experiment, to adopt the model with the incoherent tunneling because it provides the right and correct peak shape. A complete understanding of fine features of the low temperature transport justly indicated by the Reviewer, posits a nice appealing task and is a subject of the next upcoming but separate project.

At this point, the complete all-ranges-inclusive study of the full temperature dependence of the non-local thermal current in the experiment is not available. At low temperatures the thermal gradients are very small, and the non-local thermoelectric signal becomes too weak to be measured reliably. On the other hand, at high temperatures the superconductivity gets suppressed together with the CPS signal. Therefore, we have only carried out the measurements at intermediate temperatures.

We added discussion on the Coulomb blockade effects to the end of the main text. We have also added an account of the graphene nanoribbon conductance to the Methods section.

3) The small contribution of the nonlocal component with respect the local one is mainly given by the fact that the gap of the superconductor (Al film) is too near to the operating temperature. Increasing gap, the contribution of local and nonlocal EC contributions should become less important with respect CPS contribution. Is this correct? Is this progression eventually observed in some of the data taken in the experiment? Could a different superconductor such as Nb be considered?

Our response:

Increasing the gap of the superconductor indeed helps to increase the temperature gradient without suppression of superconductivity and, hence, it may enhance the non-local signal. However, the higher values of the gap may also suppress the nonlocal component in the following way. The ratio between the non-local and the local contributions is governed by the effective transmission probability of the superconductor τ_s . Theory predicts that τ_s is roughly proportional to the exponent $\exp(-d/\xi)$, where d is the distance between the dots, $\xi=(D/\Delta)^{1/2}$ is the coherence length of the superconductor, Δ is the superconducting gap, and D is the diffusion constant. It is clear from this expression that larger Δ makes ξ shorter and, with all other parameters being the same, reduces the non-local signal. Although the gap in niobium is larger, its coherence length is very small (the BCS coherence length of Al is 1600 nm while in niobium it is 40 nm) and, therefore, the overall CPS efficiency is smaller

than that in aluminum. So far, the highest CPS efficiency has been observed in aluminum. We also tried vanadium and found that the CPS efficiency using it is lower.

In order to clarify this point, we have modified the expression for the transmission probability (S22) explicitly showing the exponent $\exp(-d/\xi)$ in it.

4) Honestly, excluding Fig3a and Fig3b, the comparison between theory and experiment is not very satisfactory weak, especially for d and h. Could the author try to find the possible reasons and comment on it better?

Our response:

The experimental data have been obtained by subtracting a large background signal (local thermoelectric current). If this background is smooth and the non-local signal is sufficiently strong, the result of subtraction looks better. In Fig3c and Fig3g the background signal was not smooth enough and the non-local signal was not sufficiently strong.

The positions of the stripes for the uppermost resonance in Fig. 3c ($V_{gL} \sim -4.62$ V) seem to be slightly shifted from the expected position but the sequence is still the same: red-blue-red-blue. The apparent shift is most likely due to background subtraction that is influenced by the cross talk between the gates, in particular when the splitting efficiency is 5-10% like in the case of this resonance. The same problems that are found in Fig. 3c, are reflected directly to Fig. 3g, making the comparison with Fig. 3h challenging. However, we decided to present these data, even though the agreement between theory and experiment in Figs. 3e & 3f and Figs. 3g & 3h would have looked better without the data on this resonance. Note furthermore that in Fig. 3g we are dealing with the difference of two small currents, which is notoriously a difficult situation. For Fig. 3e, the situation is slightly better as it deals with the sum of the quantum dot currents.

In terms of theory, our main goal was to demonstrate that the theoretical model produces the same topology of the non-local signal, i.e. blue and red spots occur at the same places as in the experiment. It illustrates that the non-local thermal current originates from the product of the local conductances in agreement with Eq. (1). As for the shape and the intensity of the red and the blue spots, we could, in principle, achieve better agreement between theory and experiment in Fig. 3 by adjusting fit parameters. However, we have used the same set of parameters to fit both the local conductance peaks (see Figs. 2b and g) and the non-local signal in Fig. 3. Improving the fits in Fig. 3 would make the fits in Figs. 2b and g worse. Thus, we have chosen the parameters such that in both cases the fits look reasonably good. In particular, we have a bit enhanced the small secondary peak in Fig. 2d in order to achieve better splitting of the spots in Fig. 3.

After these clarifications I can see if their statements are sustained from the data presented. I am aware that it is a complex experiment so I do not ask extra

measurement but only other data analysis if they have it.

Let's come to minor points on the paper

a) On the bibliography I feel fair to ask a more complete reference to recent literature in particular:

concerning the thermoelectricity in superconducting and multiterminal systems probably it is fair to mention also this recent literature

"Thermoelectric properties of microstructures with four-probe versus two-probe setups" Glen D. Guttman, Eshel Ben-Jacob, and David J. Bergman Phys. Rev. B 53, 15856 (1996)

"Thermoelectric efficiency of three-terminal quantum thermal machines" Francesco Mazza et al New J. Phys. 16 085001 (2014)

"Resolving thermoelectric "paradox" in superconductors" Connor D. Shelly et al. Science Advances 2, e1501250 (2016)

"Local and nonlocal thermopower in three-terminal nanostructures" G. Michałek, et al. Phys. Rev. B 93, 235440 (2016)

"Thermopower and thermal conductance of a superconducting quantum point contact" Sergey S. Pershoguba and Leonid I. Glazman Phys. Rev. B 99, 134514 (2019)

"Nonlinear Thermoelectricity with Electron-Hole Symmetric Systems" G. Marchegiani et al. Phys. Rev. Lett. 124, 106801 (2020)

"Nonlocal Thermoelectricity in a Superconductor–Topological-Insulator–Superconductor Junction in Contact with a Normal-Metal Probe: Evidence for Helical Edge States" Gianmichele Blasi et al. Phys. Rev. Lett. 124, 227701(2020)

Our response:

We thank the Reviewer for bringing these papers to our attention. We have added all of them to the list of references and adapted the text accordingly at several places. After adding these references to the paper, we started thinking that it is also fair to include the main experimental papers on Cooper pair splitting to the manuscript: see Refs. 30-37.

b) Comment on the role of Coulomb interaction in the introduction and in some part of the main text

Our response:

Following the Reviewer's advice we now mention in the sentence after Eq. (1) that the transmission probabilities of the dots may be renormalized by Coulomb interaction. In the middle of the text we have also added a sentence "In our model we ignore Coulomb interaction because the charging energies of the dots are relatively small".

c) Tell somewhere (probably in the measurement scheme how the current is

measured, think the point the 1 questioned before). Maybe the electrical measurement scheme can be sketched in Fig.1

If the superconducting lead is kept at the same voltage of the graphene terminals tell it in the main text specifically (maybe around Eq.2).

Our response:

We have relabeled Fig. 1 to explain the main connections to the measurement setting. The relevant electrical connections and impedances are clarified in the figure caption, and detailed in the Methods section.

d) In the main text tell what is the substrate at pag4. Even if this is specified in the method and Supplementary a clarification in main text it is useful.

Our response:

We added explanation to the figure caption of Fig. 1: “..., and the silicon substrate with 280-nm-thick silicon dioxide is colored in gray”.

e) Define better the meaning of the average in Eq.3

Our response:

We added a subsection to Methods section on background subtraction.

f) In the supplementary frequently the name thermal current is used in place of the thermoelectrical current. In order to not create confusion could you check if the “thermal” before Eq12 and at the end of pag12 is correct. Thermal current is usually used to express the heat current but here I assume want to express the charge current generated by the difference of temperature (i.e thermoelectricity local or nonlocal depending on the situation). Please check also the main text

Our response:

We have replaced "thermal current" by "thermoelectrical current" everywhere in the text.

All in all, we would like to thank again Reviewer #1 for critical reading of the manuscript and numerous insightful comments that helped us to improve our presentation.

Reviewer #2 (Remarks to the Author):

Re:

Code: NCOMMS-20-18175-T

Authors: Zhenbing Tan, Antti Laitinen, Nikita Kirsanov, Alexey Galda, Mohammad Haque, Alexander Savin, Dmitry Golubev, Valerii Vinokur, Gordey Lesovik, and Pertti Hakonen
Title: Thermoelectric current in a graphene Cooper pair splitter

In the present manuscript, the authors present measurements and theoretical calculations of thermoelectric currents in a so-called Cooper pair splitter. The Cooper pair splitter is fabricated in a nanostructured device containing graphene and superconducting electrodes. The basic effect is that a superconducting injector distributes the two electrons of a Cooper pair into two, spatially separated normal metal terminals. Such a process can happen due to a purely thermal bias at zero voltage difference, as was theoretically predicted several times earlier. The authors report the measurement of the electric current due to selective heating of one of the normal sides and can control these currents using electrostatic gates. Analyzing the data the authors extract the Seebeck coefficient, and they find for the nonlocal response values as high as $250 \mu\text{V/K}$. The authors relate their findings to the creation of entangled electron pairs due to the Cooper pair splitting.

The overall impression of the manuscript is positive. The experimental results are interesting and novel and bear perspectives for the field of quantum thermodynamics. On one side, the fabricated device is very sophisticated and the measurement seems solid. The experimental signature that a finite nonlocal electric current is induced by just a temperature gradient is very clearly visible. On the other side, there are several issues in which the manuscript is not convincing. Many of those are related to the theoretical explanations, which the authors need to address before the manuscript should be considered further.

- In the very beginning and throughout the introduction, the authors emphasize the relation of Cooper pair splitting to entanglement. Now, the experiments performed for the manuscript do not allow such a conclusion since they don't test the actual entanglement of the split streams of electrons. Hence, I would recommend formulating here more carefully. Furthermore, as the authors show, the results are described by an incoherent model that neglects any quantum entanglement. One could, therefore, conclude the authors have shown that no entanglement has been produced in the present device.

Our response:

We agree that at first sight it may look surprising how the incoherent model appears as the correct model at low temperatures. This approach is to be taken with caution as discussed in detail in our Reply to Reviewer #1. The incoherent model is only used to describe the local thermoelectric current, which has complicated features that are not amenable for regular theories. Sure enough, the local phenomena cannot be employed to say anything about the entanglement.

However, our non-local phenomena are explained by the coherent transport theory. Thus, the non-local behavior does agree with theories that anticipate entanglement. The splitting efficiency deduced from our experiment is 5-10%, which is quite typical with our graphene-based devices with Al superconductor. In general, we are discussing the entanglement in a similar fashion as done in the papers where the splitting is based on electrical biasing. (see e.g. Refs. 33 and 35).

Furthermore, the plots for non-local currents obtained from the incoherent model are fundamentally different from the experimental data, which suggests that the observed non-local currents have a direct thermoelectric nature (p. 26, last paragraph). This indicates that while the experimental setup can still be subject to decoherence, which may enable the incoherent mechanism for the local thermoelectricity, the non-local current is mostly determined by the coherent electrical transport.

Following the advice of the Reviewer, we have modified a sentence on the entanglement in the conclusion paragraph to tone it a little bit down:

This work has demonstrated the possibility to use thermal gradient as *primus motor* for generating entangled electrons in graphene Cooper pair splitter.

- While the authors discuss and cite many works on the measurement of the thermopower in quantum dots, they miss in my view several important experimental works on the thermoelectric effect in superconducting heterostructures. E.g. in Kolenda et al., PRL 116, 097001 (2016) a thermopower of similar size was found in a superconductor-ferromagnet junction. A recent review of the field is found in A. Fornieri and F. Giazotto, Nat. Nanotechnol. 12, 944 (2017).

Our response:

These references are indeed relevant to our work. We have included them in the paper.

We have also included the reference “Ozaeta, A., Virtanen, P., Bergeret, F. S. & Heikkila, T. T. Predicted Very Large Thermoelectric Effect in Ferromagnet-Superconductor Junctions in the Presence of a Spin-Splitting Magnetic Field, Phys. Rev. Lett. 112, 057001 (2014)”, which is the theory paper suggesting the experiment of Kolenda et al.

- Cooper pair splitting by a thermal gradient was also studied in detail in Z. Cao, T.-F. Fang, L. Li, and H.-G. Luo, Appl Phys Lett 107, 212601 (2015). Even the model used in that paper seems to be quite similar to the one used in the present manuscript.

Our response:

We agree with the Reviewer, it is an important paper and we have already cited it in the original version of the manuscript.

- About the model, the authors claim to use scattering theory, but on the other hand

mention incoherent transport. This is somewhat contradictory. Also, a formula like $\tau_L(E)\tau_S\tau_R(-E)$ seems to represent just the product of three probabilities. How does this fit to the scattering approach? Furthermore, τ_S is calculated from the Usadel theory. Again this is not obviously compatible with the scattering approach. Note that an incoherent series addition of three elements would just add resistances, which are proportional to the inverse of the 'probabilities'.

Our response:

We use scattering theory to describe the transport through the individual quantum dots. As for the whole N - QD- S - QD - N structure, we considered two models: coherent transport model and incoherent transport model. We find that coherent model works fine everywhere except at very low temperatures. The coherent model is compatible with the Usadel equation. Indeed, it is known that after semiclassical approximation and disorder averaging the S-matrix formalism leads to the same results as the Usadel equation. Thus, the expression (S22) for the effective transmission probability of the superconductor can be derived in two ways -- either from the S-matrix or from the Usadel equation. The formula for the transmission probability of the whole structure, $\tau_L(E)\tau_S\tau_R(E)$, has such a simple form because of the two reasons. First, after disorder averaging the oscillating interference terms disappear from the transmission probability, and for a big mesoscopic structure like ours it acquires a simple classical form. Second, we assume that $\tau_S \ll 1$, therefore we neglect possible multiple reflections between the quantum dots keeping only the lowest order contribution. Thus, we omit the terms of the type $[\tau_L(E)\tau_S\tau_R(E)]^2$, $\tau_L(E)(\tau_S)^2\tau_R(E)$ etc.

We use the model of incoherent transport in order to describe the asymmetric shape of the peak in the local thermoelectric current at the lowest temperatures. We have tried several other models but could not obtain the right peak shape. We acknowledge that we do not fully understand the local transport properties of our structure at the lowest temperatures. In this regime Coulomb interaction, Kondo effect and possibly other non-trivial effects may gain importance. Incoherent transport model may be viewed as a very simple phenomenological way of taking such effects into account.

- For the determination of the Seebeck coefficient, a precise measurement of the temperature differences is necessary. However, I have the impression, the SGS thermometers are some distance away from the quantum dots. So, how exactly was the temperature calibrated? In addition, there is an issue with the determination via the SGS supercurrent, see below.

Our response:

The temperature calibration between heating-induced temperature and cryostat temperature refers to the site where the SGS junctions are located. The theoretical

formulas mentioned have no relation to the calibration; they just provide the arguments about where the thermometers have good sensitivity (see the next item).

It is true that our thermometers are located some distance apart from the quantum dots. Thus, there may be small thermal gradients along the graphene in contact with the thermometer. The gradient depends on the ratio of electron-phonon/Kapitza conductance to the substrate and the electronic conductance along the graphene. At the lowest temperatures, the electronic conductance will dominate and we may neglect the thermal gradients along graphene. At higher temperatures, thermal gradients will appear and the measured temperature difference will be an upper estimate for the temperature difference. However, if we plot the local thermoelectric data for the peak 4 in Fig. 4a as a function of temperature difference $T_L - T_S$, we find a linear dependence with $T_L - T_S$ up to the largest heating powers. We consider this as evidence that the thermal gradients are not influencing the calibration significantly at any temperatures relevant for our analysis.

We have added discussion on thermal gradients to the supplement (Note 2) and included Fig. S5 to display the linear dependence of thermoelectric current I_L on $T_L - T_S$ over a large span of temperatures.

- When describing the measurement of the local temperature, the authors use the temperature dependence of the supercurrent in the SGS structure. However, contrary to the authors' claim, the correct formula is different from the ones used in the manuscript. Namely, the dependence is rather of the form $\sim \exp(-a \sqrt{T/T^*})$ in the diffusive limit (see e.g. J. Low Temp. Phys. 106, 305 (1997) for a detailed discussion). How does this affect the measurement scheme? Furthermore, the authors calculate $E_{Th} \sim 1305 \mu\text{eV}$, which according to my estimate rather corresponds to a temperature of 13 K. If that is correct the junctions operate at a temperature much smaller than the Thouless energy and the supercurrent is basically temperature independent. Finally, what defines the 'optimum thermometer sensitivity'?

Our response:

Our theoretical remarks do not influence the temperature calibration, which is done on a fully experimental ground and which is “used as it is”. The formulas were given in order to outline the justification of why the thermometers saturate, and why they are difficult to use below 0.3 K.

The exponential form for I_c is not essential. For example, by looking at Fig. 2 in Dubos et al. Phys. Rev. B 63, 064502 (2001) one may observe that the exponential dependence is nearly correct. Furthermore, this exponential dependence has been employed in many experimental works. If there is exponential growth with lowering temperature, there needs to be saturation at some T .

The saturation analysis for our junction is complicated by the fact that the SGS junction is in the intermediate length regime where no simple theoretical expressions can be written down at present. Unfortunately, there was a typo in the Thouless energy in the original version of the manuscript. The correct value is $130 \mu\text{eV}$ which is

approximately equal to the superconducting gap. For estimates, we have employed the low temperature expression for long junctions $eRnI_c = E_{th} * b * (1 - 1.3 * \text{Exp}[-b * E_{th} / (3.2 * k_b * T)])$, and we have used b as a fitting parameter by setting $b=10.82/5$ which agrees approximately with the measured the behavior of the measured $I_c(T)$ and the $I_c R_n$ product at the lowest temperatures. The general statements about the saturation refer to this fit.

Concerning the optimum sensitivity, this is a combination of the sensitivity of $I_c(T)$ and the dependence of resistance at the gap edge on the temperature. Since the latter is more relevant for the experimental determination, the optimum is found at the lowest temperature, just before $I_c(T)$ dependence saturates.

We have shortened and modified the text in the Methods section and added clarifications to Supplementary Note 2.

- In Figure 4 a direct comparison between theory and experiment is presented. A strange feature in panel a is the "symmetry" of the current peaks in the experiment that seems to opposite to the theory. In experiment, the peaks seem to 'lean' towards the dashed line, whereas in theory they 'lean' away. This is not an essential issue, but maybe the authors have an idea, where this discrepancy comes from.

Our response:

The shape of the current peaks is sensitive to the value of the superconducting gap used in the fits. We could have made the shape of the peaks more like the experimental one if we would have chosen some smaller value of the gap. We didn't do that because of the two reasons. First, the value of the gap, $\Delta=150 \mu\text{eV}$, has been independently measured and there are no physical reasons for treating it as an additional adjustable fit parameter. Second, although by reducing the gap we could improve the shape of the peaks, the absolute value of the current would have become significantly bigger than that in the experiment. We reason that this minor discrepancy between the theory and the experiment stems from the simplifying assumption about the temperature dependence of the gap. Indeed, the model postulates the BCS gap temperature dependence with the zero temperature value of the gap $\Delta=150 \mu\text{eV}$. In reality, in the GSG structure studied in the experiment, the superconductivity in aluminum may be affected by the inverse proximity effect from graphene, and the gap in the vicinity of the dots may decrease with temperature faster than is dictated by the BCS formulas. These subtle details, however, complicate unnecessarily the description of the experiment, distracting from the fact that Fig. 4a demonstrates clearly that the proposed coherent model without any fit parameters provides a fairly good agreement with the data.

To conclude, we are grateful to Reviewer #2 for her/his constructive and thoughtful remarks that helped us to improve our presentation a lot.

Reviewer #1 (Remarks to the Author):

The authors have replied to my main concern. Still, I think that the discussion at low temperatures does not provide convincing evidence. I recognise that the authors have admitted in the paper the limitations of their explanation. I think the other pieces of evidence are sufficiently solid to guarantee a sufficient level of solidity.

Concerning my question I think they replied honestly and convincingly to issue 1), 3) and 4). I am not fully convinced by the discussion of point 2) but without further measurements, this may appear not fully clear. I suggest to the authors to attempt to better discuss the measurements done at low T in following research with a more complete report on them.

I think they replied correctly also to the point arisen by the other referee.

Concerning the paper I am ok with the revision I just leave two observations which I kindly suggest the authors to modify in the text:

a) I do not like the emphasis of this sentence "A related nontrivial phenomenon, revealed analytically in Ref. 4, was that contrary to the intuitive expectations, the superconductor can mediate transfer of heat." I think that also in Ref. 3 and 5 this point is stressed enough. It is not fair to associate this point to a single reference. The real important point of Ref4 is the fact that both Cotunnelling and CPS are treated at the same level. Something that was not done in the other two references since they consider a slightly different limits.

b) I like the addition of section C "Background subtraction" but I ask the authors to provide a more clear explanation. I had to read it two times to have an idea about what they exactly did. To be honest, still, I am not fully sure. So I ask them to consider a rephrasing making more clear what they did since it corresponds to a data treatment so it is maximal important to have very clear what they did.

Reviewer #2 (Remarks to the Author):

The authors have revised the manuscript in response to the recommendations of both referees. In my view, they have sufficiently clarified most issues raised. While I continue to assess the experiments as outstanding, the response in the theory part is not fully convincing. However, the latter issue is in some sense a matter of debate and probably will be clarified in a future scientific discussions. The manuscript should be accepted now for publication.

POINT-BY-POINT REPLIES TO REVIEWER COMMENTS

Reviewer #1 (Remarks to the Author):

R: The authors have replied to my main concern. Still, I think that the discussion at low temperatures does not provide convincing evidence. I recognise that the authors have admitted in the paper the limitations of their explanation. I think the other pieces of evidence are sufficiently solid to guarantee a sufficient level of solidity.

Concerning my question I think they replied honestly and convincingly to issue 1), 3) and 4). I am not fully convinced by the discussion of point 2) but without further measurements, this may appear not fully clear. I suggest to the authors to attempt to better discuss the measurements done at low T in following research with a more complete report on them.

A: We agree with the Reviewer's view that more experimental and theoretical work should be carried out in order to understand properly the low temperature behavior in our graphene Cooper pair splitters. We thank the reviewer for her/his encouraging suggestion, and the research clarifying these issues will be a subject of the next project and forthcoming publication.

R: I think they replied correctly also to the point arisen by the other referee.

Concerning the paper I am ok with the revision I just leave two observations which I kindly suggest the authors to modify in the text:

a) I do not like the emphasis of this sentence "A related nontrivial phenomenon, revealed analytically in Ref. 4, was that contrary to the intuitive expectations, the superconductor can mediate transfer of heat." I think that also in Ref. 3 and 5 this point is stressed enough. It is not fair to associate this point to a single reference. The real important point of Ref4 is the fact that both Cotunnelling and CPS are treated at the same level. Something that was not done in the other two references since they consider a slightly different limits.

A: We have taken the concerns of the reviewer into account and modified the text in question. To indicate slight distinction between the works, we added an additional sentence along the lines discussed by Reviewer:

"It was revealed analytically in Ref. 4 that the electric transport in the NSN structures depends on the EC process on par with the CPS. Contrary to intuitive expectations, these two processes together may enable the transfer of heat through the superconductor [3-5]."

b) I like the addition of section C "Background subtraction" but I ask the authors to provide a more clear explanation. I had to read it two times to have an idea about what they exactly did. To be honest, still, I am not fully sure. So I ask them to consider a rephrasing making more clear what they did since it corresponds to a data treatment so it is maximal important to have very clear what they did.

A: We thank the Reviewer for attentiveness to this point. Accordingly, we have rewritten Section C. We now describe the difference between the ideal background subtraction and the one used in practice. We hope that this clarifies the employed background subtraction scheme better and also makes it clear why this scheme was adopted. The related sentence in the main text has also been clarified accordingly.

Reviewer #2 (Remarks to the Author):

R: The authors have revised the manuscript in response to the recommendations of both referees. In my view, they have sufficiently clarified most issues raised. While I continue to assess the experiments as outstanding, the response in the theory part is not fully convincing. However, the latter issue is in some sense a matter of debate and probably will be clarified in a future scientific discussions. The manuscript should be accepted now for publication.

A: We are most happy that the Reviewer recommends our manuscript for publication. We appreciate her/his attentiveness to theoretical issues and encouraging comments which stimulate further research and corresponding fruitful discussions, and which will be a subject of our future work.

Reviewer #1 (Remarks to the Author):

Ok the authors has solved all the remaining issues.

The rewriteen parts are ok now.

So I support the pubblication on Nature Comm.